# Monitoring Night-Time Activity Patterns of Laying Hens in Response to Poultry Red Mite Infestations Using Night-Vision Cameras

**DOI:** 10.3390/ani15192928

**Published:** 2025-10-09

**Authors:** Sam Willems, Hanne Nijs, Nathalie Sleeckx, Tomas Norton

**Affiliations:** 1M3-BIORES Group, Division Animal and Human Health Engineering, Department of Biosystems, Catholic University of Leuven, Kasteelpark Arenberg 30, 3001 Heverlee, Belgium; sam.willems@kuleuven.be; 2Experimental Poultry Centre, Poiel 77, 2440 Geel, Belgium; hanne.nijs@provincieantwerpen.be (H.N.); nathalie.sleeckx@provincieantwerpen.be (N.S.)

**Keywords:** poultry red mite, night-time activity, computer vision, precision livestock farming, dynamic integrated pest management strategies, artificial intelligence

## Abstract

**Simple Summary:**

The poultry red mite feeds on the blood of hens at night, leading to disrupted sleep, reduced welfare, and lower productivity. Effectively managing mite infestations requires regular monitoring and the ability to adapt control strategies to farm conditions. This study investigated how the presence of mites alters hens’ night-time behaviour. Using night-vision cameras and an automated video analysis method, we quantified night-time activity patterns at both the group and individual level before and after mite introduction. Before infestation, hens displayed expected resting behaviour, with group activity declining after lights-off and remaining low until a brief anticipatory spike before lights-on. After infestation, group activity nearly doubled and remained high throughout the night, with no activity peak before lights-on. The most pronounced disruption occurred from two hours after lights-off to two hours before lights-on. At the individual level, time spent in the most active state increased from 43 min before infestation to 120 min after infestation. The increase in activity was supported by a nearly 23-fold rise in annotated behaviours linked to PRM, such as head shaking and head scratching. These findings suggest that poultry red mites disrupt the normal sleep pattern of laying hens and may induce chronic stress. Understanding the timing and extent of this disruption supports the use of continuous, automated behavioural monitoring in integrated pest management strategies in commercial egg production.

**Abstract:**

The poultry red mite (PRM) feeds on hens’ blood at night, disrupting sleep, harming welfare, and reducing productivity. Effective control may lie in dynamic Integrated Pest Management (IPM), which relies on routine monitoring and adaptation to farm conditions. This study investigated how PRM infestations affect the night-time activity of hens. Three groups of eight hens, housed in enriched cages, were monitored with night-vision cameras over a two-month period, both before and after artificial PRM introduction, while PRM levels were simultaneously recorded. To quantify changes in behaviour, we developed an activity-monitoring algorithm that extracts both group-level and individual night-time activity patterns from video recordings. Group activity between 18:00 and 03:00 was analyzed hourly, and individual activity between 21:00 and 00:00 was classified into four activity categories. Before infestation, group activity declined after 19:00, remained low from 20:00 to 01:00, and peaked just before the end of the dark period. After infestation, activity remained elevated with no anticipatory activity peak towards the end of the dark period. Individual data showed an increase in time spent in the most active activity category from 24% to 67% after infestation. The rise in calculated activity was supported by a nearly 23-fold increase in annotated PRM-related behaviours, specifically head shaking and head scratching. These findings suggest that PRM mostly disrupted sleep from two hours after lights-off to two hours before lights-on and may have acted as a chronic stressor. Automated video-based monitoring could strengthen dynamic IPM in commercial systems.

## 1. Introduction

The poultry red mite (PRM) *Dermanyssus gallinae* (De Geer, 1778) is a micropredator [1] that hides in cracks and crevices during the day and becomes active at night to feed on the blood of hens, which it needs for growth and reproduction [2,3,4,5]. It is one of the most harmful poultry pest species worldwide [6], posing significant welfare issues for laying hens. Under favourable environmental conditions, PRMs can complete their reproductive cycle in as little as seven days, leading to rapid population growth if left uncontrolled [7]. PRM is a widely spread pest species causing substantial economic losses. Prevalence rates of over 80% have frequently been reported across various European countries [8,9], with even higher rates—up to 94%—observed in the Netherlands, Germany, and Belgium [10]. The costs associated with both PRM control and production losses are estimated to exceed EUR 130 million per year across Europe [4,6]. Significant advances in PRM control will likely come through Integrated Pest Management (IPM) strategies, as suggested in the works of [6,11,12,13]. IPM is a sustainable control strategy that protects animals, humans, and the environment. It follows an eight-step process, where the second step—monitoring the pest species—is essential to its success [14].

While IPM strategies have proven effective in other agricultural sectors, PRM control in the poultry sector may require a more adaptive and dynamic approach tailored specifically to individual farm conditions [13,15]. Developing a one-size-fits-all IPM strategy for PRM is challenging due to the wide variability in factors such as seasonal temperature changes, humidity, husbandry systems, and hen breeds, all of which can affect the PRM life cycle and treatment effectiveness [15]. Consequently, economic incentives and action thresholds for PRM management differ from farm to farm. The goal, therefore, should be to establish flexible IPM strategies that can adapt to specific farm conditions. Such strategies require regular and automated monitoring to remain effective, with adjustments based on each farm’s unique circumstances and timing needs [13,15]. Addressing this need, Ref. [16] recently developed a continuous, automated monitoring approach that uses night-vision cameras to analyze hens’ behaviour for the early detection of PRM. This method shows potential to identify PRM presence through changes in activity patterns and the distribution of laying hens and also holds promise for evaluating treatment efficacy and guiding timely interventions against PRM, aligning with some of the criteria advocated by [17,18] for an efficient PRM monitoring technique.

The current gold standard for PRM monitoring relies on manual trapping tools such as cardboard traps [19] and Avivet traps [18], as automated mite counters [20] have not yet been commercialized. Moreover, approximately 20 different manual monitoring approaches have been reported [13]. However, as noted by [16], these methods are only indicative of PRM population growth because they do not capture all mites present in the poultry house [20,21]. Moreover, clustering of PRMs near traps can skew mean counts, complicating accurate population assessment [20]. Automated monitoring devices, such as the mite counter proposed by [20], can overcome some of these challenges, yet Ref. [16] highlight that it remains uncertain whether such tools can establish critical treatment thresholds, as this likely depends on both the characteristics of the farm building and the distress levels of the hens [21].

Rather than focusing solely on tools that directly monitor the pest species, monitoring the night-time activity of laying hens may serve as a valuable animal-based indicator of PRM presence and proliferation. PRM infestations are known to compromise bird health through sleep deprivation [22], increased nocturnal preening and head scratching, and elevated levels of daytime feather pecking and even cannibalism, often resulting in poor feather condition [23]. Refs. [24,25,26] manually analyzed night-time video recordings of hens and consistently reported a decrease in resting behaviour and an increase in self-directed behaviours such as preening, head scratching, head shaking, and body shaking in infested birds. Moreover, all authors reported similar day-time behavioural changes. Additional studies have linked PRM load to deteriorating feather condition. Schreiter et al. (2022) [27] showed that plumage damage increased with rising infestation levels, likely reflecting an increase in feather pecking behaviour triggered by PRM-induced irritation and stress. These findings align with clinical descriptions provided by [28], who reported that chickens heavily infested with ectoparasites exhibited itching, restlessness, loss of sleep, general weakness, loss of appetite, reduced egg production, and anemia. Similarly, in other mite species such as the northern fowl mite, who spends its life permanently on the host to feed on the blood of laying hens, Ref. [29] showed that higher levels of mite infestation were associated with hens spending less time dozing, and more time preening and being active at night. Together, these studies underscore that PRM directly disrupts rest and induces behavioural changes at night in laying hens, and these may serve as early indicators of infestation and can be exploited for monitoring within IPM programmes for effective PRM control.

To advance the development of dynamic IPM strategies, understanding the specific impact of PRM on hens’ nocturnal activity patterns remains an important knowledge gap. The primary objective of this study was to establish a night-time activity pattern for each group of eight hens both prior to (PRE-PRM) and following (POST-PRM) the artificial introduction of PRM, with the aim of identifying the hours during which PRM most disrupted sleep. A secondary objective was to determine an individual night-time activity pattern for hens during these critical hours, showing the proportion of time each hen spent in each of four activity categories, and compare these patterns under PRE-PRM and POST-PRM conditions. We hypothesize that the hens’ activity levels will differ most significantly between the two conditions three to six hours after the onset of the dark period, aligning with previous findings that suggest that PRM activity peaks three hours after the onset of the dark period [16,25,26]. Additionally, we hypothesize the baseline night-time activity level of individual hens during PRE-PRM conditions to be around 25%. This aligns with findings from EEG sleep studies, where both [30,31] utilized implanted devices to investigate sleep patterns in laying hens and established baseline wakefulness levels of 24% and 28% under undisturbed conditions. In line with this, we expect activity to rise to at least over 50% during POST-PRM conditions.

To achieve our objectives and test these hypotheses, we applied an extended version of the activity monitoring algorithm developed by [16]. The original algorithm combined Gaussian Mixture Modeling (GMM) for pixel-based motion detection [32,33] with the Segment Anything Model (SAM) [34] for individual hen segmentation, enabling pixel-based motion detection on an individual level. We extended this approach by integrating YOLOv8 [35], an object detection model, to allow the tracking of individual hens within a single night. Finally, a subset of videos was manually annotated for two behaviours (head shaking and head scratching) that are commonly associated with PRM irritation in order to provide a partial validation of the algorithm’s output and to illustrate how quantified activity patterns can be linked to biologically relevant PRM-related behaviours.

## 2. Materials and Methods

### 2.1. Animals, Housing and Husbandry

The dataset included 48 Dekalb White hens aged 18 weeks at the start of the experiment, which was conducted under a contractual agreement. The hens were housed in three enriched cages with two tiers each, accommodating eight hens per tier, within an environmentally controlled chamber (4.5 m × 6 m) The hens were housed in this chamber from 23 June 2020 to 9 November 2020, but the present study focused exclusively on data collected prior to 4 September 2020, before any PRM treatments were applied.

The hens were provided with ad libitum feed and unrestricted access to drinking water through drinking nipples, as well as access to nest boxes, perches and scratch mats. Figure 1a,b illustrate the layout of the chamber, showing the three cages and annotating features such as the tier level, cameras, feed gutter, laying nest and drinking water. Daily health and welfare assessments were performed by trained animal caretakers, with specific welfare scoring protocols implemented weekly.

The chamber temperature ranged from 20.5 °C to 28.8 °C, with relative humidity consistently above 55% throughout the study. The dark period was set from 18:00 to 03:00, with a 30 min dusk period (18:00–18:30) to allow hens to find their resting places and a 30 min dawn period (02:30–03:00) to facilitate safe waking.

### 2.2. Data Collection

The data in this study cover two distinct 16-day periods: (1) the PRE-PRM condition (2–18 July 2020, starting 10 days after habituation), when infestation levels were negligible, and (2) the POST-PRM condition (19 August–3 September 2020), which followed the deliberate introduction of PRM on 5 and 25 August 2020. A 14-day interval was maintained between the introduction of PRM and the start of the POST-PRM recordings to allow PRM populations to proliferate and establish high infestation levels in all cages. The PRE-PRM period was scheduled as early as possible after habituation to minimize the risk of PRM proliferation, thereby ensuring a clear contrast with the POST-PRM condition. Consequently, this comparison reflects hen behaviour under negligible (PRE-PRM) versus high infestation (POST-PRM) levels, rather than the possible early onset of PRM in the PRE-PRM period or the gradual proliferation of PRM during the POST-PRM period.

Weekly PRM data from cardboard [19] and Avivet traps [18] were used to confirm the difference in infestation levels between the PRE-PRM and POST-PRM conditions, as outlined in Section 2.2.1. The night-time behaviour of the hens was recorded during both periods using infrared cameras, as outlined in Section 2.2.2.

Because only two trap-based monitoring points fell within each 16-day period (PRE-PRM and POST-PRM), the PRM data could not be used for correlation analyses. Therefore, we focused on comparing hen activity and behaviour between the PRE-PRM and POST-PRM conditions, while PRM monitoring data was used for clarification and general context.

#### 2.2.1. PRM Control, Introduction and Monitoring

To ensure a negligible PRM infestation at the start of the experiment and during the PRE-PRM condition, extensive preventive control measures were taken before the hens were housed. These measures included the following: (1) immediately applying silica to all cages and the chamber after depopulation of the previous flock to kill any remaining PRM before they went back into hiding, and (2) subsequently performing dry cleaning, wet cleaning and disinfection of the chamber during the empty period. The silica treatment consisted of two separate treatments that were one week apart, in which silica was sprayed throughout the entire chamber, including walls and ceiling.

Between 23 June and 5 August 2020, no cardboard or Avivet traps were in place; instead, trained animal caretakers visually inspected the cages routinely for signs of infestation and proliferation. Despite the preventive measures, caretakers noted that eggs or PRM could still have survived or been introduced with the hens during transport, illustrating that PRM may remain undetected in well-cleaned facilities and are typically only confirmed once captured in traps. Therefore, routine cage inspection, rather than trap deployment, was considered to be an appropriate approach for assessing PRM infestation severity and confirming negligible PRM levels during the PRE-PRM period.

On 5 August 2020, PRM were deliberately introduced by distributing litter collected from a PRM-infested layer house located on the same campus as the environmentally controlled chamber onto the manure belts, where the PRM remained inaccessible to the hens. To obtain the right infestation levels for the contracted trial, a re-infestation was carried out on 25 August 2020. Prior to introduction, the manure belts were emptied and the water troughs were filled with water containing a small amount of ethanol. The infested litter was applied in small heaps on both the upper and lower manure belts of each cage (two-tier system) using a spatula and then spread evenly across the belts with a hand brush.

The exact number of PRMs introduced per cage was not quantified. Infestation levels were instead monitored weekly from the time of introduction until 9 November 2020, using two cardboard traps and two Avivet traps per cage, placed evenly across the bottom and top tiers (Figure 1c,d). After 48 h of exposure, traps were collected and stored in the freezer, and then later processed to quantify infestation levels by either counting PRMs under a microscope (cardboard traps) or weighing them (Avivet traps).

#### 2.2.2. Video Data Collection

Above each enriched cage, a night-vision camera (Foscam^®^, Shenzhen, China, model FI9912EP) was installed (Figure 1a) in order to capture top-view video recordings of the hens housed in the top tier level (Figure 2). Due to dimensional restrictions, only part of the top tier level could be covered by the field-of-view of the cameras. It was ensured that the area where the perches were located was covered by the field-of-view of the cameras. The cameras were set up to record the full dark period between 18:00 and 03:00. Cameras were connected to a network video recorder (NVR) (Dahua^®^, Hangzhou, China, model DHI-NVR4208-8P-4KS2) by means of CAT6 ethernet cables. The default camera settings were used for recording. The NVR was set up to record videos continuously at 15 frames per second with a length of 60 min at a resolution of 1280 pixels times 720 pixels.

The collected video data served two purposes: (1) all recordings were analyzed using the developed activity monitoring algorithm (see Section 2.3 and Section 2.4), and (2) a subset of videos was manually annotated for two behaviours commonly associated with PRM irritation (head shaking and head scratching) in order to provide a partial validation of the algorithm’s output and illustrate how quantified activity patterns could be linked to biologically relevant PRM-related behaviours (see Section 2.5).

### 2.3. Algorithm Development and Configuration

Figure 3, described in Section 2.3.1, presents the three stages of the proposed activity monitoring algorithm: (1) individual bird segmentation combined with YOLOv8 activity features, (2) motion detection for the movement map along with its activity features, and (3) state categorization and feature extraction.

#### 2.3.1. Activity Monitoring Algorithm

**Stage 1: Individual bird segmentation and YOLOv8 activity features.** In stage 1 of Figure 3, the algorithm applies a custom-trained YOLOv8 model (see Section 2.4.3) to each frame of the 20 s video sequence in order to detect and track each hen. This is illustrated by the coloured rectangles in each of the consecutive frames. Then, the SAM is applied in each of the coloured rectangles in the first frame of the 20 s video sequence to generate the segmentation map. Additionally, within the 20 s monitoring interval, the distance travelled by the centre-point of each detected hen is calculated. This process is then repeated until the end of the video.

**Stage 2: Detect motion for movement map and its activity features.** The second stage involves detecting the motion of the group within 20 s monitoring intervals. Each frame of the 20 s video sequence is processed using a GMM-based method to detect pixels where activity or movement occurred. All processed frames are then summed into a 20 s movement map, which is a single image where pixels with detected activity over the 20 s interval, i.e., movement pixels, are highlighted as white pixels, while areas with no activity remain black pixels. The number of movement pixels resemble the group activity within a 20 s monitoring interval. This process is then repeated until the end of the video.

**Stage 3: State categorization and feature extraction.** Stage 3 of Figure 3 introduces the concept of states, where a state represents any condition or behaviour a hen might exhibit, including inactivity. For each 20 s monitoring interval, the state of each hen is classified into one of four distinct activity categories: activity categories 1–4. The categorization process utilizes two key activity features: (1) the overlap between a uniquely segmented hen in the segmentation map and its corresponding motion blob in the movement map, hereafter referred to as the MM-OVERLAP feature, and (2) the distance travelled by the centre-point of the hen, hereafter referred to as the YOLO-DIST feature. The activity categories are defined as follows:

Category 1: MM-OVERLAP < 10% and YOLO-DIST < 10 cm

Category 2: MM-OVERLAP < 10% and YOLO-DIST ≥ 10 cm

Category 3: MM-OVERLAP ≥ 10% and YOLO-DIST < 10 cm

Category 4: MM-OVERLAP ≥ 10% and YOLO-DIST ≥ 10 cm.

We refer to Section 2.4.1 to see how the threshold of 10 cm was established and Section 4.4 for a detailed discussion on biologically relevant values of 10 cm for YOLO-DIST and 10% for MM-OVERLAP.

In activity category 1, no thresholds are exceeded, representing that little to no movement occurred within 20 s for that hen. In activity category 2, only the YOLO-DIST threshold is exceeded. In activity category 3, only the MM-OVERLAP threshold is exceeded. In activity category 4, both thresholds are exceeded, representing that movement occurred within 20 s for that hen.

Once all 20 s monitoring intervals have been analyzed, the algorithm calculates two activity features: (1) the average number of movement pixels observed across all movement maps of the video, representing the average group night-time activity level for that video, and (2) the time hens tracked for more than 50% spent in each of the four activity categories, representing the individual night-time activity pattern for that video.

#### 2.3.2. Algorithm Configuration

All developments were performed in a Python 3.10 environment on a Windows 10 laptop equipped with an Intel^®^ core i9–12900H, 2.5 GHz CPU and an NVIDIA GeForce RTX 3080 Ti GPU.

In this study, we analyzed two distinct video datasets using different configurations of the proposed algorithm. To address our primary objective of monitoring the night-time activity pattern for the three groups of hens under PRE-PRM and POST-PRM conditions, we used VIDEODATA1, comprising all 1 h videos recorded between 18:00 and 03:00. These videos were processed exclusively with stage 2 of the proposed algorithm to calculate the average number of movement pixels across all movement maps. For each hour in VIDEODATA1, the average number of movement pixels across all movement maps was extracted as a feature to represent hourly group activity.

To address our secondary objective of determining individual night-time activity patterns under PRE-PRM and POST-PRM conditions, we used VIDEODATA2, a dataset comprising 1 h videos recorded between 21:00 and 00:00. These videos were processed using the full algorithm, which was adapted to track individual hens across the videos, allowing for a maximum tracking duration of 180 min per night (21:00 to 00:00). We calculated the time each hen spent in the four defined activity categories and the duration for which each hen was tracked each night between 21:00 and 00:00. To ensure accuracy, only hens tracked for at least 50% of the night (21:00 to 00:00) were included in the calculation of time spent in each activity category.

The simplified algorithm implementation to analyze VIDEODATA1 was driven by computational constraints, excluding neural networks for hen segmentation and tracking (stage 1) as well as event detection (stage 3), which significantly reduced processing time. Table 1 shows the algorithm configurations and highlights the key differences between VIDEODATA1 and VIDEODATA2.

### 2.4. Algorithm Implementation

#### 2.4.1. Camera Calibration

To calibrate the cameras for each cage, a reference distance of 20 cm was marked with a red line on a single frame from each camera (see example for cage 1 in Figure 2). This reference distance was then measured in pixels for each camera to determine the pixel-to-centimetre conversion factor. For cage 1, the 20 cm reference corresponded to 120 pixels, resulting in a scale of 6 pixels per centimetre. For cage 2, the same distance corresponded to 100 pixels, resulting in 5 pixels per centimetre, and for cage 3, it was 105 pixels, resulting in 5.25 pixels per centimetre. These calibration values were used to ensure consistent spatial measurements across all cages. As such, the 10 cm threshold for the YOLO-DIST feature used in state categorization was standardized for all cameras.

#### 2.4.2. Motion Detection Parameters

Willems et al. (2025) [16] analyzed two datasets with varying video quality in their paper. The first dataset lacked fine details, such as the contours of individual feathers on the body of a hen, while the second dataset, captured with optimized camera settings, provided higher-quality videos where these finer details were clearly visible. In the present study, the same motion detection parameter values as those used for processing high-quality videos in [16] were applied.

#### 2.4.3. Custom-Trained YOLOv8 Detection Model for Tracking

A total of 1002 images were extracted from the three cameras, with each hen manually labelled using a rectangle, also referred to as a bounding box (Figure 2). The bounding boxes were annotated using the LabelImg software (version 1.8.6) [36]. The dataset was divided into 701 images for training, 200 for validation, and 101 for testing. Training was conducted for up to 500 epochs with early stopping enabled if performance did not increase within the last 50 epochs, using a batch size of 16 and the AdamW optimizer (Ultralytics YOLOv8, version 8.2.71). During training, the model was optimized using three complementary loss functions: the Complete Intersection over Union (CIoU) loss, which evaluates how well the predicted bounding boxes overlap with the ground truth and thus guides box regression; the Distribution Focal Loss (DFL), which refines bounding box localization by modelling the coordinates as probability distributions; and the Varifocal Loss (VFL), which improves classification by making the predicted confidence scores reflect both the correctness of the class label and the quality of the bounding box. Model performance on the validation and test sets was evaluated using the mean Average Precision (mAP) averaged over Intersection over Union (IoU) thresholds from 0.5 to 0.95 (mAP@50–95) and the mAP at IoU of 50% (mAP@50). Once the detection model was trained, YOLOv8’s built-in ByteTrack [37] (Ultralytics YOLOv8, version 8.2.71), an advanced tracking algorithm inspired by SORT [38] and DeepSORT [39], was used to accurately track hens across videos in VIDEODATA2. Because the hens were not individually marked with tags or colours, tracking relied entirely on YOLOv8’s built-in ByteTrack. As a result, birds could only be followed within a single night, and it was not possible to verify whether the same individuals were tracked across different nights, which is also common in commercial aviary systems. Since only part of the cage was covered by the camera (see Section 2.2.2), it is very likely, though not certain, that some of the same individuals were monitored across nights. However, without a reliable method of identifying them across nights, the statistical analyses were subject to certain limitations, as outlined in Section 4.7.

A one-minute example video of YOLOv8 tracking with ByteTrack is available at https://osf.io/uxzgj/ (created on 1 August 2025), accompanied by a brief description of the video.

### 2.5. Behaviour Annotation of Videos

To establish a link between algorithm-derived activity patterns and biologically relevant PRM-related behaviours, a subset of videos of cage 2 was manually annotated. For both the PRE-PRM and POST-PRM conditions, videos recorded between 22:00 and 23:00 were selected for annotation.

Two specific behaviours commonly associated with PRM irritation were annotated: head shaking and head scratching. Annotation was performed using a custom Python script that allowed observers to click directly on a hen at the moment a target behaviour occurred. A behaviour was included only if it met the following criteria: 

(1) The behaviour matched a clear definition. Head shaking was defined as rapid turning of the head from side to side, covering about 180 degrees, with the head held above the shoulders. Head scratching was defined as scratching the head with a foot (adapted from [25]).

(2) To avoid overcounting, repeated instances of the same behaviour within a two-second interval were not annotated again.

The annotation work was conducted as part of a student project involving five trained students. Before annotation, all students received structured training covering PRM biology, welfare impacts, and video examples of the two target behaviours, as well as practical instructions on using the annotation script and applying the inclusion and exclusion criteria. Each student then annotated an assigned batch of videos independently.

After this first round, annotations were assessed and finally selected by a trained observer (S.W.) for consistency and accuracy.

### 2.6. Visualizations and Statistical Analyses

#### 2.6.1. PRM Monitoring Data

As outlined in Section 2.2, PRM monitoring data served only to provide clarification and general context. Hence, the PRM population growth during the POST-PRM period was visualized using weekly data from cardboard and Avivet traps using the JMP^®^ Pro software (version 17.0.0).

#### 2.6.2. Hen Activity and Behaviour Monitoring

The average hourly group activity between 22:00 and 23:00, as calculated by the algorithm, was visualized alongside the manually annotated counts of head shaking and head scratching using the JMP^®^ Pro software. This comparison was based on seven consecutive days selected near the end of both the PRE-PRM and POST-PRM periods.

The results of VIDEODATA1 were visualized using JMP^®^ Pro software. Additionally, for VIDEODATA2, also using JMP^®^ Pro software, a linear mixed model (standard least squares with restricted maximum likelihood estimation) was applied to investigate the interaction effect between the study period (PRE-PRM and POST-PRM) and cage number (CAGE 1, CAGE2, CAGE3). The time hens spent in activity category 1, 2, 3 and 4 served as the dependent variables. The main fixed effect was the study period (PRE-PRM or POST-PRM, α = 0.05), modelling systematic differences in activity between the two periods. Cage number (CAGE 1, CAGE2, CAGE3, α = 0.05) was included as an additional fixed effect to account for systematic variability between cages. The interaction effect between study period and cage number was included to investigate differences among cages during the PRE-PRM and POST-PRM conditions. Study day was modelled as a random factor to capture variability between different nights, while hen ID nested within cage number was included as a random effect to account for variability among individual hens observed within each cage on a given night.

## 3. Results

### 3.1. PRM Monitoring

During the PRE-PRM condition, animal caretakers observed no visible signs of PRM infestation or proliferation in any of the three cages. Because no traps were deployed in this period, negligible infestation levels were inferred from the combination of extensive preventive measures prior to housing and routine cage inspections.

During the POST-PRM condition, PRM was monitored using cardboard and Avivet traps. Figure 4 shows the trap data upon its removal over four consecutive weeks (*x*-axis), with the last three weeks (19 August, 26 August, and 2 September) corresponding to the POST-PRM period. The *y*-axis is split into three panes, representing cage 1 (top), cage 2 (middle), and cage 3 (bottom). Within each pane, mean PRM counts from the two cardboard traps (#, shown in blue) and mean weights from the two Avivet traps (mg, shown in red) are displayed. By 2 September 2020, all three cages had reached high PRM infestation levels, with cage 3 exhibiting the highest infestation.

### 3.2. Hen Detection Model and Hen Tracking for VIDEODATA2

Figure 5 shows that during training (top row) and validation (bottom row) of the hen detection model, the box loss (first column), the class loss (second column), and the distributed focal loss (third column) decreased steadily without overfitting the model, while performance metrics on the validation set, indicated by precision, recall, mAP@50, and mAP@50–95, increased. The model stopped training at epoch 270, with the best fitness observed at epoch 220. On the validation set, the model achieved a mAP@50–95 of 0.8896 and a mAP@50 of 0.9910. On the test set, the model achieved a mAP@50–95 of 0.8861 and a mAP@50 of 0.9927.

Table 2 presents average tracking results per cage for VIDEODATA2 (21:00–00:00) during PRE-PRM (16 days) and POST-PRM (16 days) conditions. The average number of detected hens within the field-of-view of the cameras ranged between 4.5 and 5.9, while the average number of hens that were tracked for more than 50% of the time ranged between 3.1 and 4.5. When a hen was tracked for more than 50% of the time, the average time it was tracked for ranged between 144.4 and 157.69 min, as shown in the bottom row of Table 2.

### 3.3. Group Night-Time Activity Pattern (VIDEODATA 1)

Figure 6 shows the hourly group activity during PRE-PRM (red) and POST-PRM (blue) periods. Solid coloured lines connected by dots represent the average activity per hour, while shaded box plots illustrate the distribution of movement data within each hour. To clarify, each night consisted of nine 1 h videos, and 16 nights were analyzed for both the PRE-PRM and POST-PRM periods. Within each video, movement was quantified every 20 s and summarized into a movement map. For each hour of the night (*x*-axis), the box plots present the distribution of movement pixels across all 20 s movement maps from all 16 nights, while the solid lines connected by dots show the average number of movement pixels for that hour. The left pane in Figure 6 corresponds to cage 1, the middle pane to cage 2, and the right pane to cage 3.

As shown in Figure 6, during POST-PRM conditions, the activity level remains relatively constant and consistently higher compared to PRE-PRM conditions for cage 1, 2 and 3, as shown by the solid blue lines. In contrast, during PRE-PRM conditions, the activity level is high at the beginning and end of the night, with a noticeable decline in activity during the hours in between. For all three cages, the most pronounced differences occur during the hours of 20:00, 21:00, 22:00, 23:00, 00:00, and 02:00, as highlighted by the non-overlapping box plots for these specific hours.

### 3.4. Individual Night-Time Activity Pattern (VIDEODATA 2)

Least squares mean values, together with their standard deviations, are presented in Table 3, where superscript letters indicate statistically significant differences within a particular activity category. Table 3 shows that the average time spent in activity categories 1, 2 and 3 decreases by 89% (from 47 to 5 min), 71% (from 53 to 15 min) and 48% (from 8 to 4 min), respectively, when comparing PRE-PRM and POST-PRM conditions. Similarly, the average time spent in activity category 4 increases by 192% (from 43 to 120 min).

Overall, hens spent the least time in activity category 3, indicating that the observed states where the YOLO-DIST threshold was not exceeded and the MM-OVERLAP threshold was exceeded were the least likely to occur. This means that it was rare for the distance travelled by the centre-point to be less than 10 cm while the overlap between the movement map and the segmentation map exceeded 10%.

The linear mixed model showed statistically significant differences (*p* < 0.05) between all cages during PRE-PRM and POST-PRM conditions regarding the time hens spent in activity categories 1, 2, 3 and 4, with the exception of cage 2 for activity category 3. While no significant differences were reported between cages within the PRE-PRM or POST-PRM conditions for activity category 1, significant differences were reported between cages for activity categories 2, 3 and 4 in PRE-PRM conditions, reflecting variability in baseline night-time activity levels.

### 3.5. Comparison Between Activity and Behaviour

Figure 7 compares calculated activity and annotated behaviours for cage 2 during the hours 22:00–23:00 for seven consecutive days near the end of the PRE-PRM (10–16 July) and POST-PRM (27 August–2 September) periods. The primary *y*-axis displays the number of annotated events as a stacked bar plot (blue = head shaking, red = head scratching). The secondary *y*-axis shows the average hourly group activity, quantified by the algorithm, represented by a black line with solid dots. Both datasets were restricted to the same one-hour window, enabling direct comparison between quantified activity and observed behaviours.

In the PRE-PRM period, very few events were observed, with an average of 5.0 annotated behaviours (3.3 head shakes and 1.7 head scratches). In contrast, the POST-PRM period showed a marked increase, with an average of 114.3 annotated behaviours (65.2 head shakes and 49.2 head scratches), representing a nearly 23-fold increase. A similar, though less pronounced, trend was observed for average hourly group activity, which increased by a factor of 3.4 compared with the PRE-PRM period (21,543 vs. 72,755 movement pixels on average).

## 4. Discussion

Section 4.1.1 discusses PRM infestation levels in the PRE-PRM and POST-PRM periods, while Section 4.1.2 compares automated and manual PRM monitoring techniques and highlights the challenge in interpreting and comparing both approaches. Section 4.2, Section 4.3 and Section 4.4 discuss the outcomes of the activity monitoring algorithm in more detail. Section 4.5 provides details on comparing the outcome of the activity monitoring algorithm with behavioural annotations, while Section 4.6 and Section 4.7 provide perspectives on future research and applications, as well as limitations.

### 4.1. PRM Monitoring

#### 4.1.1. Comparison of PRM Infestation Levels in PRE-PRM and POST-PRM

During the PRE-PRM period, no cardboard or Avivet traps were deployed, and infestation levels were instead assessed through routine caretaker inspections of the cages. These inspections revealed no visible signs of PRM infestation or proliferation in any of the three cages, consistent with the extensive preventive measures implemented prior to housing. Specifically, silica was applied twice throughout the entire chamber—including walls and ceiling—alongside dry cleaning, wet cleaning, and disinfection during the empty period. Such measures are known to reduce residual PRM populations [13]. In addition, a 10-month monitoring project on 20 commercial farms, as cited by [13], reported that combining dry and wet cleaning with treatments such as silica can significantly delay reinfestation [40]. Together, these findings support the interpretation that PRM pressure was negligible during the PRE-PRM stage.

Nevertheless, it cannot be fully excluded that residual PRM, eggs, or newly introduced individuals (e.g., carried during hen transport) were present at low levels. This limitation reflects a broader challenge in PRM management, even in thoroughly cleaned and treated facilities, infestations may persist unnoticed until they are detected by traps or have reached levels that produce visible signs. In this context, relying on routine inspections rather than traps offers a practical approach, providing reasonable assurance that infestations remain negligible while acknowledging that complete PRM-free conditions are difficult to guarantee, particularly in facilities with a known history of infestation. Thus, the PRE-PRM period should be interpreted as representing minimal rather than complete absence of PRM pressure.

Following the deliberate introduction of PRM on 5 and 25 August 2020, weekly monitoring with cardboard and Avivet traps confirmed the successful establishment of high infestations across all cages during the POST-PRM period. Both trap types showed clear increases in PRM numbers. As shown in Figure 4, PRM levels were broadly consistent across cages, although cage 3 reached the highest trap values. Infestation levels remained high throughout the 16-day POST-PRM period, indicating that the hens were continuously exposed to the PRM challenge in all cages.

#### 4.1.2. Comparison of Automated and Manual PRM Monitoring

Unlike studies on the northern fowl mite, where hens can be inoculated with a precise number of mites to investigate their effect on the night-time activity of laying hens [29], true baseline PRM levels could not be established in our study, as outlined in Section 4.1.1. Moreover, trap-based methods, such as cardboard and Avivet traps, only provide estimates of PRM infestation and are known to underestimate population size, with results affected by trap placement and sampling frequency [17,20,21]. Automated PRM counters [20] reduce some of these limitations but may not reliably determine treatment thresholds, as these depend on building characteristics and hen distress levels [21].

In contrast to direct methods such as traps and automated mite counters, a camera-based approach monitors the behavioural impact of PRM on hens, offering a broader perspective on how infestations affect bird welfare. Both manual and automated approaches face challenges in achieving accurate PRM monitoring. While traps and automated counters provide direct measures of PRM numbers, our method assesses their impact on hen activity and behaviour, which makes interpretation different and direct comparisons between methods more complex [16].

### 4.2. Hen Tracking for VIDEODATA2

Table 2 shows the consistent long-term tracking of hens in all cages throughout PRE-PRM and POST-PRM conditions. This consistency reflects the hens’ tendency to remain in their chosen sleeping spots throughout the dark period, as reported by [16], likely due to their poor night vision [41]. While the present study involved small groups of hens, under conditions with a much lower stocking density than typically found in commercial housing systems, our earlier work [16] demonstrated the feasibility of individual detection in semi-commercial conditions with over 50 hens within the camera’s field-of-view. Together, these findings suggest that consistent individual tracking may be achievable in larger flocks, although the practical challenges—such as occlusions, camera placement, and movement dynamics—become more pronounced in commercial settings. Such individualized monitoring is crucial for implementing dynamic Integrated Pest Management (IPM) strategies, allowing timely interventions to combat PRM. This aligns with observations from [42], who reported the persistence and long-term establishment capacity of PRM. Effective management requires rigorous cleaning, consistent and continuous monitoring, and preventive measures to mitigate infestations and improve welfare and productivity.

### 4.3. Group Night-Time Activity Pattern (VIDEODATA 1)

Figure 6 illustrates the hourly night-time activity patterns under PRE-PRM and POST-PRM conditions during the dark period (18:00 to 03:00). During the POST-PRM condition, group activity remained relatively constant and consistently higher compared to PRE-PRM conditions throughout the night. In contrast, under PRE-PRM conditions, activity decreased after 19:00, stabilized at lower levels from 20:00 to 01:00, and sharply increased in anticipation of the lights-on period between 02:00 and 03:00. These results suggest that PRMs disrupt sleep patterns during the night, particularly between 20:00 and 01:00. The observed disruption aligns with findings by [16,25,26], who reported (although without providing evidence for such claims) that PRM activity peaks approximately three hours after the onset of darkness. However, this pattern contrasts with the observations of [5], who found PRM activity to peak 5–11 h after darkness onset under a 12 h light–dark cycle. This contrast further highlights the need for a PRM monitoring approach tailored to farm-specific conditions.

While our findings partially support our hypothesis that significant behavioural differences emerge three to six hours after lights-off, the pronounced activity spike observed between 02:00 and 03:00 under PRE-PRM conditions was unexpected. The absence of this peak during the POST-PRM period suggests that PRM may have acted as a chronic stressor, leading to sleep disruption and suppressing anticipatory activity before the lights-on phase. This interpretation is in line with [30], who showed—using implanted EEG devices—that prolonged heat exposure in laying hens elevated night-time wakefulness. However, while the observed changes are consistent with a PRM-related effect, caution is warranted. The time gap of approximately one month between the PRE- and POST-PRM recordings raises the possibility that other factors may also have contributed to the altered activity patterns, as discussed in more detail in Section 4.7.

### 4.4. Individual Night-Time Activity Pattern (VIDEODATA 2)

Refs. [30,31] utilized implanted EEG devices to investigate sleep patterns in laying hens, establishing baseline wakefulness levels of 24% and 28%, respectively, under undisturbed conditions. The statistically significant findings in Table 3 align with the observations of [30]. During the PRE-PRM condition, hens in all three cages spent an average of 43 min in activity category 4, corresponding to a baseline night-time activity level of 24% (relative to the maximum tracking time of 180 min). In contrast, under the POST-PRM condition, the average time spent in activity category 4 increased to 120 min, reflecting a night-time activity level of 67%. This prolonged wakefulness under PRM disturbances is comparable to the heightened activity observed in hens experiencing prolonged temperature-induced stress, as described in [30]. These results underscore how persistent stressors—whether environmental or parasitic—disrupt rest throughout the night. Our findings support our hypothesis of a baseline night-time activity level of around 25% [30,31], which rises to over 50% when hens are disturbed by PRM. It is also noteworthy that the linear mixed modelling showed statistically significant differences between cages during the PRE-PRM condition, showing variations in baseline night-time activity levels. This highlights the importance of using a robust monitoring technique capable of handling individual and group-level differences in night-time activity patterns.

Table 3 shows that hens spent the least time in activity category 3, indicating that it was uncommon to observe a state where the hen’s centre-point moved very little (YOLO-DIST < 10) while the overlap between the movement map and segmentation map exceeded 10% (MM-OVERLAP > 10). Most of the observed 20 s intervals fell into one of three other categories: both thresholds being exceeded (activity category 4), neither threshold being exceeded (activity category 1), or only the YOLO-DIST threshold being exceeded (activity category 2). This pattern is expected, as 20 s is a relatively long time for a hen to remain stationary (YOLO-DIST < 10) while also maintaining a high overlap (MM-OVERLAP > 10). The threshold values used in this study—10 cm for YOLO-DIST and 10% for MM-OVERLAP—were carefully chosen to be biologically meaningful: (1) complete inactivity represents no movement at all (YOLO-DIST = 0) and no overlap between the movement map and segmentation map (MM-OVERLAP = 0), (2) a centre-point travelled distance of 10 cm over 20 s is very small, capturing even the subtlest movements indicative of activity, and (3) an overlap of 10% ensures the algorithm detects even minimal overlap between the movement map and segmentation map.

These thresholds were chosen to define activity category 1 as the range representing states between complete inactivity (YOLO-DIST = 0 and MM-OVERLAP = 0) and very minimal movement (YOLO-DIST < 10 and MM-OVERLAP < 10), effectively capturing all instances from complete inactivity to subtle activity. We argue that this approach can be adapted to other scenarios by adjusting the threshold values, as described in more detail in Section 4.5.

### 4.5. Comparison Between Activity and Behaviour

To partially validate the algorithm output, quantified activity was compared with manually annotated behaviours known to be associated with PRM irritation. For comparability, seven consecutive days near the end of both the PRE-PRM and POST-PRM periods were visualized (Figure 7). Although not a full validation dataset, this approach confirmed that at least part of the variation in quantified activity could be attributed to biologically relevant behaviours. The large increases in activity and behaviour shown in Figure 7 are consistent with previously reported behavioural changes under PRM infestation.

Grooming behaviour, which consists of preening and scratching [43], is critical for defence against ectoparasites. Preening is the primary defensive response, allowing birds to remove or damage ectoparasites on their feathers and skin, while scratching with the feet targets areas that cannot be reached through self-preening, such as the head [44,45]. Head shaking has been described as an alerting response that reflects heightened attention to environmental stimuli [46]. Temple et al. (2020) [25] further interpreted night-time head shaking in infested hens as anticipation of negative events, linking this behaviour to compromised welfare in less-preferred environments [47,48]. Studying hens on commercial farms, Temple et al. (2020) [25] reported significant reductions in night-time activity, preening, head scratching, and head shaking after acaricidal treatment. Similarly, Petersen et al. (2021) [26], working on commercial aviaries and free-range farms, showed that PRM infestations significantly increased night-time head shaking, head scratching, preening, body shaking, and vertical wing shaking, while reducing resting behaviour, with all measures improving again after effective treatment. Additionally, Kilpinen et al. (2005) [24] also reported behavioural changes, as the PRM-infected hens showed higher night-time self-grooming and head scratching. Maurer & Fölsch (1993) [49] further noted that hens abandoned their preferred sleeping locations on perches when PRM infestation increased, reinforcing the evidence that PRM pressure disrupts normal night-time behaviour.

While studies of [24,25,26] manually observed night-time video recordings, Ref. [16] demonstrated that such changes can also be detected automatically by monitoring night-time activity with video cameras, providing a scalable approach for flock-level PRM surveillance. Similarly, the findings of the current study reinforce that both activity levels and behavioural information are promising animal-based indicators for PRM monitoring.

Figure 7 shows that annotated behaviours increased almost 23-fold between PRE-PRM and POST-PRM, while average hourly group activity increased 3.4-fold. This difference in fold increases can be explained by the spatial and temporal resolution of the two approaches: behaviours were annotated at the individual level every 2 s, whereas activity was calculated for the whole group in 20 s intervals, with movement summed into a single movement map per interval. Because the algorithm evaluates group movement in 20 s windows, it does not distinguish between multiple events occurring within the same interval. For example, if a hen performs several head shakes while remaining in the same location, a human annotator records each event (up to once every 2 s), whereas the algorithm’s GMM-based motion detection simply registers repeated activation of the same pixels, resulting in no increase in calculated activity. Likewise, the YOLO-based classification applies fixed thresholds that do not account for variation in intensity. This explains why activity measures can stabilize at a plateau once infestation reaches high levels and hens repeatedly show activity within the same 20 s interval, in contrast to individual behavioural monitoring at a finer temporal resolution (e.g., 2 s). This difference accounts for the much larger fold increase observed in behavioural annotation compared to activity monitoring.

### 4.6. Future Research and Applications

#### 4.6.1. Commercial Implementation and Welfare Relevance

Future research should prioritize implementing these camera-based PRM monitoring approaches in commercially relevant settings, as the current experimental setup only involved groups of eight hens. Deploying multiple cameras, or moveable cameras mounted on a railing system, to monitor all hens in larger flocks could provide farm-specific, animal-based insights into PRM presence and distribution, thereby facilitating more targeted and effective interventions within dynamic IPM strategies. The present study compared a period with negligible PRM infestation (PRE-PRM) to one with high infestation (POST-PRM), separated by a one-month interval. As shown in Figure 7, the gap between the PRE-PRM period (ending 18 July 2020) and the POST-PRM period (starting 19 August 2020) leaves room for the early onset of PRM in PRE-PRM or increasing proliferation of PRM in POST-PRM. Future studies should therefore focus on this interval for the early detection of PRM infestation and proliferation, now that the main hypotheses and research questions addressed here have been answered. As PRMs likely act as a chronic stressor and may disrupt or eliminate sleep, incorporating sleep monitoring into welfare assessment protocols could provide a valuable metric for evaluating hen well-being. As Putyora et al. (2023) [31] emphasize, welfare assessments typically focus on daytime conditions, whereas the night-time environment and its impact on sleep are frequently overlooked, despite the clear requirements for sleep in vertebrate species.

#### 4.6.2. Flexible and Complementary Monitoring Approaches

The approach developed in this study, which applies biologically meaningful thresholds to define activity categories, is flexible and can be adapted to detect specific behaviours by adjusting thresholds and monitoring parameters. For instance, to detect dust bathing—a stationary behaviour with localized movement—thresholds such as YOLO-DIST = 10 cm and MM-OVERLAP = 90% could serve as a biologically relevant starting point. Shortening the monitoring interval (e.g., from 20 s to 2 s) would likely enhance detection accuracy. This adaptability allows the algorithm not only to identify specific behaviours but also to capture them as short video sequences that can be integrated into video classification algorithms for broader behavioural monitoring applications.

Activity monitoring effectively reveals overall differences in night-time activity associated with PRMs, but behavioural monitoring of PRM-specific behaviours provides a more fine-grained approach suitable for early PRM detection. Group-level activity captures total movement, including non-PRM-related behaviours, which may obscure infestation effects. By contrast, behaviours such as head shaking and head scratching directly reflect PRM irritation and may therefore offer greater sensitivity for detecting infestation and tailoring advice within dynamic IPM programmes. Monitoring at a finer temporal scale (e.g., every two seconds) also avoids the plateau effect observed in activity measures (as outlined in Section 4.5) and enables the detection of more subtle changes linked to PRM. Activity and behavioural monitoring should thus be regarded as complementary: activity analysis offers a robust framework for identifying active individuals and generating relevant video sequences of their behaviour, which can then be classified into PRM-related and non-PRM-related behaviours for more precise monitoring.

#### 4.6.3. Towards a Practical Computer Vision Monitoring System

In Willems et al. (2025) [16], the authors showed how night-time location monitoring can be used to pinpoint areas in which PRMs are most likely to be present. Building on this work, the current study introduces activity monitoring, which could generate short video sequences for behavioural classification at the individual level. Taken together, these advances support the development of a comprehensive monitoring system that (1) integrates location, activity, and behaviour monitoring, (2) combines measurements at both group and individual levels, and (3) addresses practical challenges encountered in commercial settings.

To realize such a system, we propose a framework that integrates GMM for pixel-based motion detection, YOLO for detecting and tracking individual birds’ movement and location, and SAM for the segmentation of birds. This combination enables the localized classification of activity into four biologically meaningful levels while distinguishing true hen activity from artefacts caused by environmental factors, while also allowing short video sequences of behaviour to be generated.

A clear example of such an artefact is sprinkler activation at night [16]. In this case, GMM detects extensive pixel changes caused by water spraying across the field-of-view, whereas YOLO, which tracks the centre-point movements of birds, does not register activity. When most birds in the video exceed the GMM threshold but not the YOLO threshold within a 20 s interval, the system could use this information to classify this as false activity. In contrast, sudden disturbances such as a loud noise or a bird falling from a perch will awaken the majority of birds in the field-of-view [16], causing both GMM and YOLO to be triggered simultaneously for the majority of the birds. This information can be used to classify these events as group movements unrelated to PRM presence. Incorporating such logic enables the algorithm to filter out false positives caused by environmental artefacts (e.g., sprinkler systems, spider webs, occlusions, loud noise, bird falling of perch). This improves robustness and strengthens the feasibility of applying computer vision-based PRM detection in real-world commercial applications.

Ultimately, an approach that integrates GMM, YOLO and SAM could reliably identify the birds in the stable that are most active and display the highest levels of PRM-related behaviours, thereby providing farm-specific insights and enabling targeted PRM control.

While a single camera can capture part of the flock at night, multiple cameras, or moveable cameras mounted on a railing system, would allow a more comprehensive overview of the entire flock. A practical computer vision-based PLF tool for PRM monitoring should capture a sufficiently large proportion of the flock to enable robust statistical analyses and targeted PRM control. Moreover, monitoring only night-time activity may not be cost-effective for end-users such as farmers. Future research should therefore focus on integrating the proposed night-time monitoring framework with day-time monitoring, using the same computer vision methods. If multiple behaviours or welfare issues can be addressed with a single camera system, this would provide a more integrated view of hen welfare while maintaining costs for the farmer, since one system could serve multiple purposes.

### 4.7. Study Limitations

In this study, only part of the cages could be recorded due to dimensionality restrictions, which may have resulted in some behaviours and movements outside the recorded areas being missed. Future studies could overcome this limitation by using multiple cameras or wider-angle lenses to expand the recorded area and capture the activity of all hens, thereby enhancing the robustness of statistical analyses and interpretations. Additionally, as outlined in Section 4.6, future research should focus on translating the proposed methods to commercially relevant settings to increase generalizability and external validity.

Additionally, we used a linear mixed modelling approach to test for statistically significant differences. This method normally requires that the same individuals can be identified and measured repeatedly over time, assuming that these repeated measurements are dependent. However, we were unable to identify and follow the same hen across nights, meaning that this assumption could not be fully met. Instead, we assumed that hens react in a broadly similar way to PRM infestation in terms of their night-time activity. Despite this limitation, we still chose to apply the linear mixed model because it is a widely used method for analyzing repeated measures data. We report the results for completeness and transparency, even though the model’s assumptions were not fully satisfied. Consequently, statistically significant differences should be interpreted with care.

It is also important to consider the temporal gap of approximately one month between the PRE- and POST-PRM recordings. Although changes in activity patterns were observed following PRM infestation, this gap introduces the potential for confounding influences, such as age effects or microclimatic changes, which may also have contributed to differences in night-time activity.

Another limitation relates to the manual annotation of behaviours. Although annotations by five trained students were assessed and selected by a single trained observer (S.W.) for consistency and accuracy, some subjectivity in behavioural labelling and selection cannot be excluded. This is a common challenge when relying on manual observations, even when structured training and clear definitions are applied.

Finally, differences between manual and automated PRM monitoring approaches should be acknowledged, as outlined in Section 4.1. Both manual and automated approaches provide different types of information and face distinct limitations, complicating interpretation and direct comparison [16].

## 5. Conclusions

This study quantified group- and individual-level night-time activity patterns of hens before and after poultry red mite (PRM) infestation. The largest disruptions occurred between 20:00 and 01:00, supporting our hypothesis that PRM primarily affects hens three to six hours after lights-off. An activity spike between 02:00 and 03:00 under PRE-PRM conditions was absent in POST-PRM, suggesting suppression of the normal circadian rise in activity. At the individual level, baseline activity increased from around 24% to over 67% in the most active category, confirming our hypothesis of a rise to over 50% under infestation. Together, these findings indicate that PRM acted as a chronic stressor, with variability across cages highlighting the need for farm-specific monitoring. These findings underline how knowledge of the timing and extent of PRM-related behavioural changes can enhance the use of automated, continuous behavioural monitoring to support dynamic Integrated Pest Management (IPM) strategies for effective PRM control.

Comparison between manually annotated behaviours and calculated activity confirmed that both methods captured biologically meaningful changes that are promising animal-based indicators for PRM monitoring. The larger fold increase in behavioural annotation reflected its finer temporal and spatial resolution, making it a promising indicator for early PRM detection. By contrast, group activity monitoring provided a broader flock-level perspective, revealing hourly differences in the night-time activity pattern associated with PRM, though its measures may reach a plateau once infestation reaches higher intensities.

Future applications should prioritize commercial implementation, using multiple or moveable cameras to monitor entire flocks and generate farm-specific, animal-based insights. Integrating GMM, YOLO, and SAM allows localized classification of hen activity while filtering environmental artefacts, and activity monitoring further enables the identification of PRM-related behaviours for targeted PRM control. Extending these methods to daytime monitoring could provide a more comprehensive view of hen welfare while enhancing cost-effectiveness and farmer adoption. While this study provides valuable insights, an inability to follow the same hens across nights and the one-month gap between recordings introduce limitations. These should be considered when interpreting the results.

## Figures and Tables

**Figure 1 animals-15-02928-f001:**
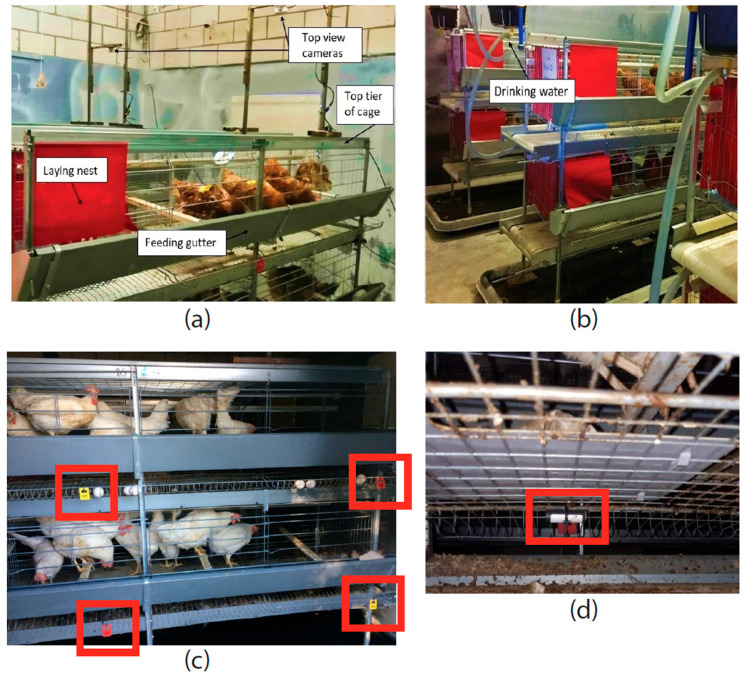
Overview of the environmentally controlled chamber and the three cages: (**a**) highlights the top tier of the cage together with top-view cameras; (**b**) shows the drinking system; (**c**) shows the location of the poultry red mite (PRM) monitoring traps in red rectangles, where red tags represent Avivet traps and yellow tags represent cardboard traps (see Section 2.2.1); (**d**) shows the placement of a trap. Additionally, (**a**,**b**) show the setup used in previous experiments, hence the brown birds housed in the cages, while (**c**,**d**) show pictures of the present study, housing white birds.

**Figure 2 animals-15-02928-f002:**
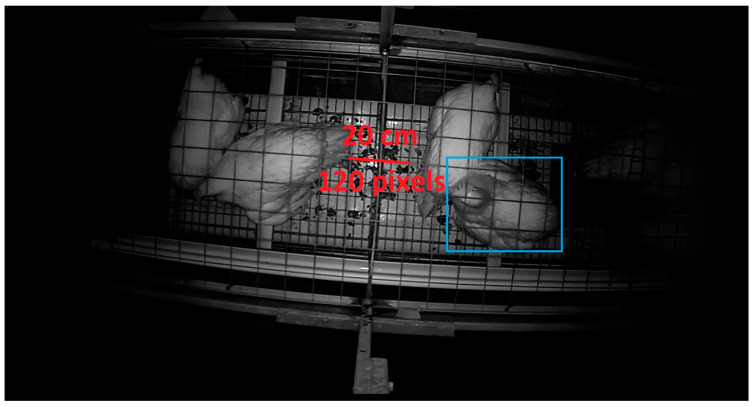
Example of the field-of-view of the top-view camera for cage 1. The central red line represents a reference distance of 20 cm, corresponding to 120 pixels, as described in Section 2.4.1. The light blue rectangle shows an example of the bounding box labelling process outlined in Section 2.4.3.

**Figure 3 animals-15-02928-f003:**
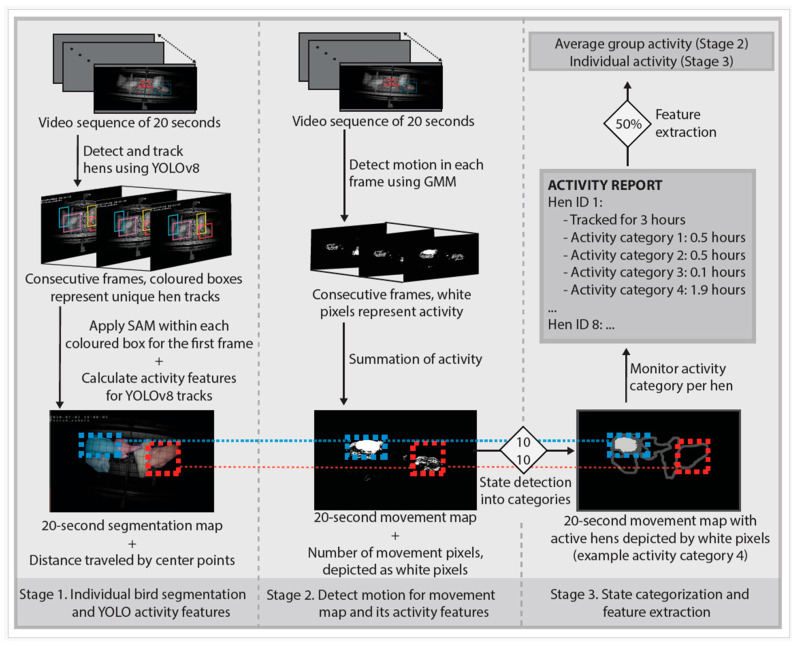
Overview of the activity monitoring algorithm. Black solid arrows indicate the flow of analysis within and between stages. The blue and red dashed rectangles, connected by dashed straight lines at the bottom of Figure 3, illustrate how the overlap between the segmentation map in stage 1 and the movement map in stage 2 is used to define events. These overlaps are essential for state categorization in stage 3 of the algorithm. A 10% overlap threshold combined with a minimum distance of 10 cm travelled by each hen’s centre-point for state categorization is used, represented by the diamond shape at the bottom of Figure 3. For stage 3, an example of an active hen is shown in the movement map (depicted by white pixels), which would be categorized into activity category 4, i.e., both thresholds are exceeded. At the end of each video, an activity report is generated to compute specific activity features. Only hens tracked for more than 50% of the video are included to calculate average activity features, as shown by the diamond shape at the top of Figure 3 in stage 3.

**Figure 4 animals-15-02928-f004:**
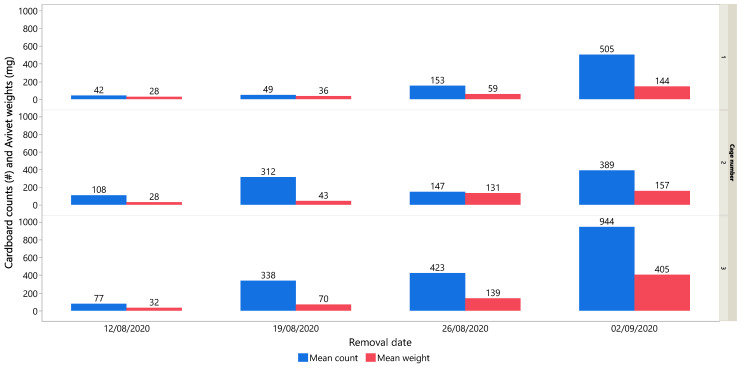
PRM trap monitoring data. Blue colours show data for mean cardboard PRM counts (# in the *y*-axis label denotes the number of PRMs found in the traps), while red colours show data for mean Avivet weights (measured in milligrams). The top pane corresponds to cage 1, the middle pane to cage 2, and the bottom pane to cage 3.

**Figure 5 animals-15-02928-f005:**
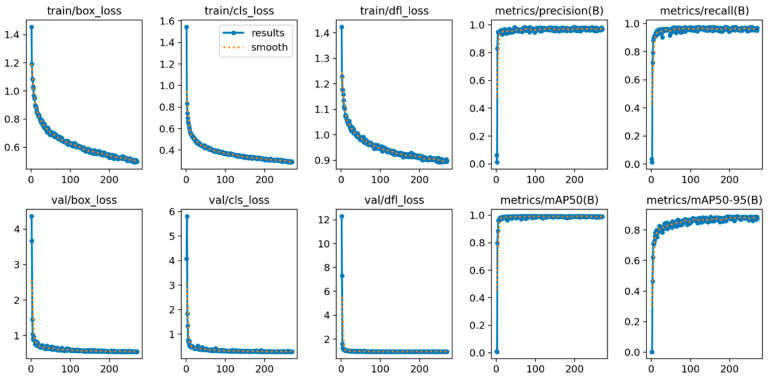
Overview of training and validation of the object detection model. The top row includes the training losses for box, class, and distribution focal loss, while the bottom row presents the corresponding validation losses. The remaining plots display the validation performance metrics (precision, recall, mAP50, and mAP50–95). Each subplot illustrates the respective loss or performance metric on the *y*-axis against the number of epochs on the *x*-axis. The solid blue lines indicate the raw values, while the dashed orange lines represent the smoothed values.

**Figure 6 animals-15-02928-f006:**
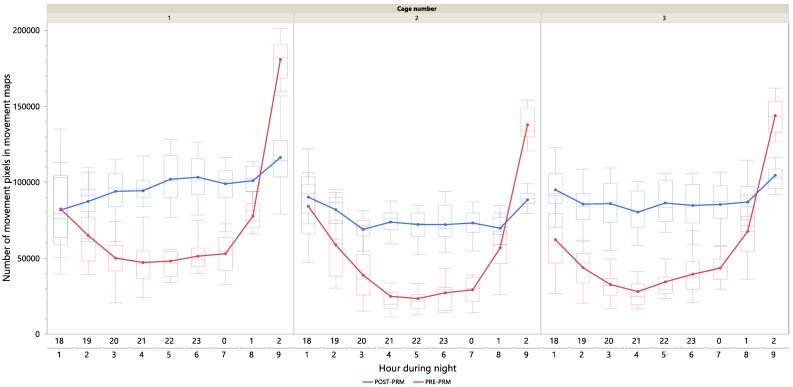
Group night-time activity pattern (VIDEODATA1). The *y*-axis represents the hourly group activity expressed by the number of movement pixels across all movement maps. The *x*-axis represents the specific hour during the night (18:00 to 03:00 or 1 h to 9 h). The *x*-axis is grouped into each of the three cages, where the left pane corresponds to cage 1, the middle pane to cage 2, and the right pane to cage 3. Red colours represent PRE-PRM condition and blue colours POST-PRM condition. Boxplots are shaded and represent a comparison between the data distribution of PRE-PRM and POST-PRM condition for a specific hour, while solid coloured lines connect the average hourly values.

**Figure 7 animals-15-02928-f007:**
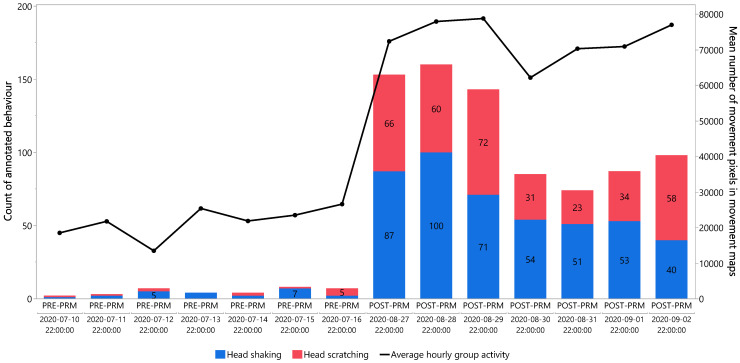
Comparison of activity and behaviour for cage 2 for PRE-PRM and POST-PRM. Counts of annotated behaviours are displayed on the primary *y*-axis, while average hourly group activity is plotted on the secondary *y*-axis, expressed as the mean number of movement pixels in movement maps. Behavioural counts are plotted as a stacked bar plot, where blue bars represent head shaking counts and red bars represent head scratching counts. In the middle of the coloured bars, the exact number of annotated behaviours is displayed in black (only for counts ≥ 5).

**Table 1 animals-15-02928-t001:** Comparison between VIDEODATA1 and VIDEODATA2.

Characteristics	VIDEODATA1	VIDEODATA2
Time range	18:00 to 03:00	21:00 to 00:00
Paper objective	Primary objective	Secondary objective
Algorithm configuration	Algorithm Stage 2	Full algorithm, adapted to track individual hens across videos
Feature calculated	Average number of movement pixels across all movement maps	Time each hen spent in each activity category
Feature representation	Average hourly group activity	Nightly individual activity
Maximum tracking time	Not applicable	3 h per night (21:00 to 00:00)
Hens included in order to calculate feature	All hens within the field-of-view of the camera	Hens tracked at least 50% of the night (90 min)
Reason for simplified implementation	Driven by computational constraints; excluded neural networks (Stage 1) and event detection (Stage 3)	Not applicable

**Table 2 animals-15-02928-t002:** Tracking results per cage for VIDEODATA2 during PRE-PRM and POST-PRM conditions.

Metric	Cage 1		Cage 2		Cage 3	
	PRE-PRM	POST-PRM	PRE-PRM	POST-PRM	PRE-PRM	POST-PRM
Average number of detected hens	5.6	4.5	5.5	5.5	5.9	5.9
Average number of hens tracked > 50% of the time	4.5	3.1	4.3	3.4	4.1	3.8
Average tracking time (min) per hen that was tracked > 50% of the time	154.37	157.69	144.40	155.07	146.51	152.96

**Table 3 animals-15-02928-t003:** Individual night-time activity pattern. Average time (min) hens spent per activity category per cage, shown as least squares means ± standard deviations for the PRE-PRM and POST-PRM periods.

Activity Category	Cage	PRE-PRM	POST-PRM	Difference Between PRE- and POST-PRM
1	1	42.3 ± 2.6 ^a^	6.0 ± 3.2 ^b^	−86%
	2	49.4 ± 2.6 ^a^	6.0 ± 2.9 ^b^	−88%
	3	48.5 ± 2.8 ^a^	3.3 ± 2.8 ^b^	−93%
Average		46.73	5.10	−89%
2	1	47.4 ± 2.9 ^c^	18.5 ± 3.4 ^b^	−61%
	2	65.4 ± 2.8 ^a^	16.7 ± 3.2 ^b^	−74%
	3	47.4 ± 3.0 ^c^	11.0 ± 3.0 ^b^	−77%
Average		53.40	15.40	−70.67%
3	1	10.2 ± 0.7 ^a^	3.2 ± 0.9 ^b^	−69%
	2	5.1 ± 0.7 ^b^	3.8 ± 0.8 ^b^	−25%
	3	8.6 ± 0.8 ^a^	4.2 ± 0.8 ^b^	−51%
Average		7.97	3.73	−48.33%
4	1	54.2 ± 3.9 ^a^	121.7 ± 4.7 ^b^	+125%
	2	30.5 ± 3.9 ^c^	114.6 ± 4.2 ^b^	+276%
	3	45.2 ± 4.2 ^a,c^	124.5 ± 4.1 ^b^	+175%
Average		43.30	120.27	+192%

Within each activity category, levels marked by different superscript letters indicate significant differences specific to that category.

## Data Availability

The datasets generated for both group-level and individual night-time activity and behaviour, as well as PRM trap data, are publicly available in the Open Science Framework repository at the following link: https://osf.io/uxzgj/ (created on 1 August 2025).

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
