# Peer review of "Monitoring Night-Time Activity Patterns of Laying Hens in Response to Poultry Red Mite Infestations Using Night-Vision Cameras"

_animals, 2025, doi:10.3390/ani15192928_

Round 1
Reviewer 1 Report
Comments and Suggestions for Authors
Review of “Monitoring night-time activity patterns of laying hens in response to poultry red mite infestations using night-vision cameras.”
In this manuscript the authors track the nighttime activity of birds before and after infestation with poultry red mite. The poultry red mite (PRM) is a blood feeding ectoparasite of chickens that lives in the environment and travels to the birds at night to blood feed. The authors use video recordings and machine learning to track group bird activity and individual bird activity at night following a mite infestation.
As the manuscript is currently written, it reads more to an engineering audience than a biology audience. The emphasis of this paper is on the machine learning/video camera aspect, and so this study may be better suited to a ‘technology in agriculture’ or other focused journal. There is very little description of what behaviors are being performed, instead relying on “categories” of activity that mean nothing to the reader about what or how much movement is being performed by the chickens.
The authors need to expand their methods so the manuscript can be fully evaluated. As written, it is difficult to evaluate how this adds to the existing literature in the field. There are already several papers evaluating behavior, including at night, of chickens when exposed to red mite infestation. Therefore, the novelty of this manuscript may be in the methods of behavior evaluation rather than the effects of PRM on chicken behavior.
Another concern is that the authors did not monitor for mites consistently throughout the study (no Avivet traps were placed pre-infestation) and their statement of a “negligible PRM infestation” is not substantiated with data. There is no description of how mites were introduced, or how many mites were introduced to the cages, and the report of mites post infestation is at only one time, despite data being collected over 16 days; there could be a different level of mite infestation in each cage at the beginning of that period compared to the end of that period because mite populations grow so quickly. The size of the mite population at different data collection points is an important part of this study, and is necessary to assess what these bird responses mean (are they actually caused by PRM? There is no mite-free control, so these could be age effects). One claim the authors make is that this tool could be used as part of an IPM program, and to implement such a program a (mite population) threshold through monitoring would need to be established which would (likely) be related to chicken behavior/restlessness at night.
The authors have failed to cite any literature on chicken behavior responses to other blood-feeding (or feather-feeding) mites or ectoparasities, including nighttime activity (e.g., Jacobs et al. 2019, Vezzoli et al. 2015, Murillo et al. 2020, 2024 - you could also look at other animal/parasite systems e.g. Smythe et al. 2015)
Methods:
The authors should explain further how the algorithm was validated (is the algorithm available for public/research use?), and how individual birds were marked and tracked.
Report weekly mite counts per cage. Does trapping remove mites from the experiment/significantly impact the overall mite population?
Results:
Data were collected over a 16 day period - was this averaged for each cage (e.g. FIgure 5)? Please be more clear in the text.
(line 400) “Cardboard trap counts and Avivets weights showed no large differences between the cages, indicating a consistent infestation rate across all three cages.” These data are not shown and no statistics validate this statement.
Discussion:
(line 442) “other factors, such as seasonal changes or environmental variation, may also have contributed to the altered activity patterns” - Weren’t the birds in an environmentally controlled house?
(line 485) “Future research should prioritize implementing these PRM monitoring approaches in commercially relevant settings.” What monitoring approach are the authors referring to? There is no linking of the behavior responses recorded to actual mite numbers recorded. While they may have the data to support this statement, it is not presented in the current manuscript.
(line 489) this is a bit of a throwaway about REM cycles - elaborate on what is expected in chickens/provide a citation
Reviewer 2 Report
Comments and Suggestions for Authors
The manuscript describes an interesting approach to poultry welfare. Sleep behaviour assessment in red mite infestations has been reported in previous studies, but the authors presented a different evaluation method.
The observations of this manuscript are minimal
L 34- Please include the duration of the experiment
L 45- Please consider including AI or computer vision model in the keywords
L 165- How were you able to determine it was not the same bird being recorded if they were able to move?
Reviewer 3 Report
Comments and Suggestions for Authors
The manuscript addresses an important and timely topic in poultry science, namely the impact of Dermanyssus gallinae infestations on laying hen welfare and behaviour, assessed through automated video analysis. The study is well-structured, clearly written, and provides valuable insights into the disruption of nocturnal activity caused by poultry red mites (PRM). The integration of advanced computer vision methods with behavioural monitoring is innovative and holds promise for precision livestock farming and dynamic Integrated Pest Management (IPM) strategies. Overall, the manuscript is of good quality and contributes to both animal welfare science and applied pest management.
While the manuscript is well-structured and provides valuable insights, some revisions are recommended to improve its scientific rigor and clarity; detailed suggestions are provided below:
-
Sample size and generalizability:
The study uses three groups of eight hens in controlled housing conditions. While the proof-of-concept is valuable, extrapolation to commercial farm settings with thousands of hens should be made more cautiously. The authors acknowledge this in part, but the limitations section could further emphasize how group size and housing type may affect external validity. -
Temporal gap between conditions:
The one-month interval between PRE-PRM and POST-PRM recordings introduces possible confounding factors (e.g., seasonal variation, microclimatic changes). Although acknowledged, the impact of these confounders on nocturnal activity patterns warrants deeper consideration. -
Statistical model assumptions:
The linear mixed model assumes repeated measures of the same individuals, but hens were not consistently identifiable across nights. While justified, this limitation weakens the strength of the statistical inference and should be explicitly mentioned in the conclusions as well. -
Figures 1–5 are generally clear, but adding higher-resolution images or supplementary video material would improve comprehension, particularly regarding hen tracking.
-
Some references (e.g., to recent EEG studies or PRM epidemiology) could be expanded for completeness.
-
The discussion occasionally repeats points made earlier in the introduction; slight streamlining would improve readability.
